# Evaluation of Effects of Ractopamine on Cardiovascular, Respiratory, and Locomotory Physiology in Animal Model Zebrafish Larvae

**DOI:** 10.3390/cells10092449

**Published:** 2021-09-17

**Authors:** Kumail Abbas, Ferry Saputra, Michael Edbert Suryanto, Yu-Heng Lai, Jong-Chin Huang, Wen-Hao Yu, Kelvin H.-C. Chen, Ying-Ting Lin, Chung-Der Hsiao

**Affiliations:** 1Department of Bioscience Technology, Chung Yuan Christian University, Chung-Li, Taoyuan City 3020314, Taiwan; g10965007@cycu.edu.tw (K.A.); g10865013@cycu.edu.tw (F.S.); g10865014@cycu.edu.tw (M.E.S.); 2Department of Chemistry, Chinese Culture University, Taipei 11114, Taiwan; LYH21@ulive.pccu.edu.tw; 3Department of Applied Chemistry, National Pingtung University, Pingtung 900391, Taiwan; hjc@mail.nptu.edu.tw; 4Department of Biotechnology, College of Life Science, Kaohsiung Medical University, Kaohsiung 80708, Taiwan; u108552001@kmu.edu.tw; 5Drug Development and Value Creation Research Center, Kaohsiung Medical University, Kaohsiung 80708, Taiwan; 6Department of Chemistry, Chung Yuan Christian University, Chung-Li, Taoyuan City 320314, Taiwan; 7Center for Nanotechnology, Chung Yuan Christian University, Chung-Li, Taoyuan City 320314, Taiwan; 8Research Center for Aquatic Toxicology and Pharmacology, Chung Yuan Christian University, Chung-Li, Taoyuan City 320314, Taiwan

**Keywords:** ractopamine, zebrafish, cardiovascular physiology, locomotion, molecular docking, homology modeling, propranolol, rescue effect

## Abstract

Ractopamine (RAC) is a beta-adrenoceptor agonist that is used to promote lean and increased food conversion efficiency in livestock. This compound has been considered to be causing behavioral and physiological alterations in livestock like pig. Few studies have addressed the potential non-target effect of RAC in aquatic animals. In this study, we aimed to explore the potential physiological response after acute RAC exposure in zebrafish by evaluating multiple endpoints like locomotor activity, oxygen consumption, and cardiovascular performance. Zebrafish larvae were subjected to waterborne RAC exposure at 0.1, 1, 2, 4, or 8 ppm for 24 h, and the corresponding cardiovascular, respiratory, and locomotion activities were monitored and quantified. In addition, we also performed in silico molecular docking for RAC with 10 zebrafish endogenous β-adrenergic receptors to elucidate the potential acting mechanism of RAC. Results show RAC administration can significantly boost locomotor activity, cardiac performance, oxygen consumption, and blood flow rate, but without affecting the cardiac rhythm regularity in zebrafish embryos. Based on structure-based flexible molecular docking, RAC display similar binding affinity to all ten subtypes of endogenous β-adrenergic receptors, from *adra1aa* to *adra2db*, which are equivalent to the human one. This result suggests RAC might act as high potency and broad spectrum β-adrenergic receptors agonist on boosting the locomotor activity, cardiac performance, and oxygen consumption in zebrafish. To validate our results, we co-incubated a well-known β-blocker of propranolol (PROP) with RAC. PROP exposure tends to minimize the locomotor hyperactivity, high oxygen consumption, and cardiac rate in zebrafish larvae. In silico structure-based molecular simulation and binding affinity tests show PROP has an overall lower binding affinity than RAC. Taken together, our studies provide solid in vivo evidence to support that RAC plays crucial roles on modulating cardiovascular, respiratory, and locomotory physiology in zebrafish for the first time. In addition, the versatile functions of RAC as β-agonist possibly mediated via receptor competition with PROP as β-antagonist.

## 1. Introduction

The excessive usage of drugs in veterinary medicine affects non-targeted aquatic animals, which are also effecting human population through drinking water [1,2]. These drugs are excreted through manure, which is being used as fertilizers, thus polluting aquatic life as well [3]. One widely used drug is ractopamine (RAC), a synthetic β-adrenergic agonist, used as feed additive for increased effectiveness of feed, increased growth and muscle leanness, also termed as ‘lean meat agent’ [4,5]. β-adrenergic receptors are classified into β1, β2, and β3. β1- and β2-adrenergic receptors being most abundant and expressed in heart, kidney, lungs, and blood vessels, while β3-adrenergic receptors are found in the adipose tissue and a very few in the heart [6]. Adrenergic receptors help in modulating several metabolic functions in fish regarding oxygen uptake, cardiac rate, vascular resistance, and hemoglobin–oxygen binding affinity [7,8].

RAC modulates the metabolism and redirects the nutrients to muscles from the adipose tissue, increases lipolysis and protein synthesis [9,10]. The U.S. Food and Drug Administration (FDA) authorized RAC in 2000, and it has been used in selected food animals as feed additive in different countries worldwide. Although RAC is currently banned in 160 countries, including the European Union members [11] and Taiwan [12], use in livestock is still in practice in some countries, including USA, Canada, Japan, South Africa, and Mexico [13].

Over 88% of ingested RAC is excreted through animal feces [14], that is disposed of without treatment and becomes contaminated in subsurface, brooks, rivers, and other territorial waters [15]. Different concentrations of RAC were detected in groundwater near piggery (0.054 μg/L) [15], and agricultural catchment area (1.3 × 10^−5^ to 5.4 × 10^−4^ μg/L) [16]. RAC was also detected in sewerage ponds near pig farms in variable amounts ranging from 0.134 to 0.524 μg/L [15] and 0.138 μg/L to 30 μg/L [17]. In a previous investigative study, in hospital sewage and rivers in Taiwan, four β-agonists were detected, while 70% of the accumulated specimens were RAC [18].

However, recent studies demonstrated that the use of RAC may affect heart, thyroid, urethra, and prostate [19], and also causes cardiac toxicity [20]. Few studies have been performed on RAC exposure to non-target aquatic animals. It has been revealed that chronic exposure of 44 days, disrupts the endocrine system in Japanese Medaka (*Oryzias latipes*), exposed to different doses of RAC between 5 and 625 μg/L [21]. While chronic exposure of different concentrations of RAC in adult zebrafish (*Danio rerio*) at 0.1, 0.2, 0.85, 8.5, and 85 μg/L induced behavioral changes and oxidative stress [22], and increases the heart rate of zebrafish embryos and larvae [23].

In silico structure-based molecular simulation imparts a great and helpful method to demonstrate a possible ligand–receptor binding location. Due to tremendous advancements in computational scope and efficiency in recent years, molecular docking has become a valuable and accessible method to show the molecular interactivity between the ligand and its receptor at atomic resolution [24]. This method is exceptionally crucial for the academic community in drug discovery and toxicity forecasting. The consequences of the technique can come up with advice on searching out drugs and designing new concepts with more significant binding tendencies [25]. To see how RAC can bind to zebrafish endogenous β-adrenergic receptors, we employed a dry lab molecular simulation to anticipate further reliable receptor structure supporting their ligand–receptor interactions at an atomic scale. To construct a 3D receptor structure model, firstly, we fetched the required sequence from UniProt (https://www.uniprot.org/; accessed on 9 March 2021) and then spotted the most comparative corresponding structure as a template. By doing the sequence alignment between the receptor and template, homology modeling [26] builds up 3D protein structure models to see whether RAC can bind to the constructed receptor. We optimized the RAC structure, explored for probable binding cavities of the receptor, and performed molecular docking for the ligand–receptor binding situation.

Aquatic and environmental toxicity has been a great concern for the last decade. Due to excessive globalization and industrial growth, aquatic life is being affected indirectly. The use of zebrafish as an animal model has wide advantages, including transparency of embryo and larva, high throughput examination, short experimental duration, cost-effective, genetic resemblance with humans (around 85%), less quantity of compound is required, and it is also been approved by the U.S FDA and European Medical Agency (EMEA) for harmful and safety evaluation for Investigative New Drug (IND) compliance [27]. The objective of this study is to explore the potential physiological impacts of the feed additive RAC that is being excessively used in livestock and aquatic industry as a repartitioning agent, which in turn causes some physiological and behavioral alterations in non-target animals and ultimately humans. Although RAC is banned in different countries, its misuse is still a big concern, especially in Taiwan.

## 2. Materials and Methods

### 2.1. Housing and Breeding of Zebrafish

AB strain of zebrafish stock were acquired from the Taiwan Zebrafish Core Facility at Academia Sinica (TZCAS) and housed in laboratory for one month prior to the experiment under recirculating water supply at 26.5 °C and a 10/14 dark–light cycle. We fed artemia and commercial feed to fish twice. Female to male ratio in accordance to 1:2 were kept in a breeding tank for the duration of a night by separating them with a glassy separator following our previous published study [28,29]. The barrier was removed the next morning. After collection and sanitization, the embryos, were then put down at 28 °C in methylene blue water. Until the third day of post-fertilization, all the expired larvae were being taken out and methylene blue water medium was replaced daily. All of these zebrafish experimental protocols and principles were authorized by the Chung Yuan Christian University animal care and welfare committee (Number: CYCU109001, issue date: 20 January 2020).

### 2.2. Chemical Preparation

RAC was purchased from ANPEL Laboratory Technologies Inc., Shanghai, China and was prepared as 1000 ppm for stock solution using ddH_2_O. Then stock solution was further diluted into desired concentrations using ddH_2_O at the time of exposure. PROP was purchased from Sigma-Aldrich, Inc., St Louis, Missouri, USA with >99% purity and 1000 ppm stock solution was prepared using ddH_2_O, which then was further diluted into desired concentrations using ddH_2_O.

### 2.3. Automated Behavioral Analysis Using ZebraBox

In this study, we used a high-throughput monitoring enclosure named ‘ZebraBox’ developed by Viewpoint Company (ViewPoint 3.22.3.85, ViewPoint Life Sciences, Inc., Civrieux, France, 2014 (http://www.viewpointlifesciences.com; accessed on 9 June 2021) to analyze zebrafish behaviors. After 96 h post-fertilization (hpf), 48 zebrafish larvae were collected in a 9-cm petri dish for each control and RAC treated groups. The number of experimental animals was based on prior studies [29]. Zebrafish were exposed to RAC at 0.1, 1, 2, 4, and 8 ppm concentrations and were labelled as the treatment group and a control group kept under the same conditions for a 24-h incubation period. After the 24-h incubation period, each group of zebrafish larvae were transferred to 48-well plate for locomotion quantification. Acclimation time of almost 1 h was given before the 48-well plate was kept on the ZebraBox machine plate. After placing the 48-well plate in the ZebraBox, additionally a 10-min acclimation time before proceeding the procedure was given.

Video tracking analysis was performed in a 48-well plate individually for each group of both zebrafish under the infrared illuminating system of the ZebraBox, which has the ability of high-quality imaging in dark. Video was recorded for 80 min (10 min of light and dark period alternatively) for each group and the same video was then used for Rotation and Burst analysis. Overall locomotion activity was analyzed on the basis of total distance travelled, burst count, and rotation count. Movement categorization as cruising (normal speed), increased body activity (short, powerful, and intermittent activity), and no activity (freeze activity) was made in the previous study [30]. As per our previous protocol [31], we categorized the velocities in three ranges to measure the total distance chronology as: velocities exceeding 20 mm/s were named larger or increased body activity, 0.5 to 20 mm/s was called as cruising activity, and less than 0.5 mm/s as inactivity. Moreover, for total burst count, the simulation of video track parameters was set as burst 20 pixels/s and freeze (no movement). Furthermore, rotation chronology was determined by the clockwise and anti-clockwise movements in millimeters and contemplated a diameter over 2 mm as one rotation and rotation less than 2 mm was abandoned. Moreover, the back angle was set at 60° [29]. The tests were performed in duplicate.

### 2.4. Zebrafish Larvae Cardiac Physiology and Rhythm Assay

Zebrafish embryos were collected after 48 hpf and expose to RAC for 24 h at different concentration (0.1, 4, and 8 ppm). To keep the position of the larvae, 3% methylcellulose was used right before recording as the mounting medium. Recording was done using a high-speed digital charged coupling device (CCD) (AZ Instrument Corp., Taichung City, Taiwan) mounted on to an inverted microscope (ICX41, Sunny Optical Technology, Zhejiang, China) while a Hoffmann modulation contrast objective lens with 40× magnification and LPlans objective lens with 10× magnification was used to record videos with high resolution and contrast. HiBest Viewer software package (AZ Instrument, Taichung City, Taiwan) was deployed to record at a frame rate of 200 frames per second (fps) for 10 s. Ventricle movement, focused on the cardiac physiology parameters, was recorded using Hoffman modulation contrast objective lens, while LPlans objective lens was used to check the rhythm of the heart chamber. To check the variability of zebrafish heartbeat, Poincare plot plug-in from Origin Pro 2019 software (OriginLab Corporation, Northampton, MA, USA) was used to calculate the sd1 and sd2 of both heart chamber. All cardiac performance and rhythm assay was calculated according to our previously published protocol [32].

### 2.5. Zebrafish Oxygen Consumption Analysis

The oxygen consumption analysis was conducted using a 24-well plate kept in the Sensor Dish Reader (SDR) and software MicroResp® version 1.0.4 (Loligo Systems, Viborg, Denmark) was used to record the oxygen concentration in the well during experiment time. Approximately 46 zebrafish larvae at 96 hpf were exposed to RAC for 24 h. At experiment time, the larvae were put together with 80 μL of the compound. One zebrafish larva was put on each well, except one well was used as a blank and filled with ddH_2_O only. The oxygen consumption rate was recorded for 80 min for each group of zebrafish larvae and the experiment was done in triplicate.

### 2.6. Blood Flow Measurement

The blood flow measurement was done according to our previously published study reported by Santoso et al. [33]. Briefly, about 10 zebrafish larvae were placed under an inverted microscope (ICX41, Sunny Optical Technology, Zhejiang, China) in a 5-cm plastic petri dish and mounted with 3% methylcellulose to prevent animal movement. After that, Hoffmann modulation contrast objective Lens with 40× magnification was used to visualize the blood vessel in dorsal aorta (DA). High-speed CCD camera (AZ Instrument, Taichung City, Taiwan) capable to record at 200 fps was used to record the blood movement for 10 s. The video was further analyzed using “Trackmate” plugin at ImageJ Version 1.53e to get the position of blood cells in every frame [34,35]. The data then processed in Microsoft Excel (Excel version in Microsoft Office Professional Plus 2016 for Windows) in to get the velocity of detected blood cell. The experiment was done in duplication.

### 2.7. Molecular Docking Analysis

To simulate the binding between the inspected RAC and PROP with ten different subtypes of zebrafish endogenous β-adrenergic receptors, we performed homology modeling and molecular docking in Windows 7 Home Basic on an Asus computer (Intel^®^ CoreTM i7 2.67 GHz processor), using Swiss Model web-server (https://swissmodel.expasy.org/; accessed on 14 March 2021) [36] and Modeller Software v9.20 [37] and flexible CDOCKER [38,39]. Homology modeling first constructs ten three-dimensional protein structures using the sequences of zebrafish endogenous β-adrenergic receptors from UniProt (https://www.uniprot.org; accessed on 9 March 2021) and the template homologous crystal templates from Protein Data Bank (PDB, https://www.rcsb.org; accessed on 9 March 2021). The related protein sequences for adra1aa, adra1ab, adra1ba, adra1bb, adra1d, adra2a, adra2b, adra2c, adra2da, and adra2db are used to locate four homologous crystal templates, 6kux (Crystal structures of the alpha2A adrenergic receptor in complex with an antagonist RSC), 5v54 (Crystal structure of 5-HT1B receptor in complex with methiothepin), 7e32 (Serotonin 1D (5-HT1D) receptor-Gi protein complex), and 7e2z (Aripiprazole-bound serotonin 1A (5-HT1A) receptor-Gi protein complex) by Swiss Model tools. Among four crystal binding sites, we overlapped RAC with the four crystal ligands with the identities of 85.3% (6kux), 79.9% (5v54), 73.7% (7e32), 79.8% (7e2z), and getting the best fit with the 6kux ligand. Thus, 6kux was chosen as the template structure. Best predicting structure models were identified by the two scoring functions, DOPE [40], and GA341 [41]. Flexible molecular docking then did the simulation of the binding between RAC or PROP with β-adrenergic receptors of zebrafish. Note that the 6kux (adra2a, *Homo sapiens*) binding site was also taken for the flexible docking as comparison. A cavity searching method, eraser algorithm [42], is used to locate the best binding pocket. The partial charges of all atoms within the receptor were determined, and all hydrogens were restored by CHARMm force field [43]. Molecular docking results showing the binding between RAC or PROP with zebrafish endogenous β-adrenergic receptors were reported in the Table 1 of results section.

### 2.8. Co-incubation of RAC and PROP Experiment

Based on the results, RAC exposure could induce the locomotion, cardiac physiology, respiratory performance, and blood flow velocity alterations in larval zebrafish. Additionally, based on molecular docking analysis, it showed that RAC can interact with β-adrenergic receptors with high affinity. To validate this hypothesis, we evaluated the physiological outcomes of zebrafish after co-incubating with RAC and β-adrenergic receptor antagonist of PROP. For all rescue tests (excluding oxygen consumption assay), the highest concentration of RAC, 8 ppm, was selected and co-incubated with 1 or 4 ppm of PROP. The same protocols were applied as before with 24-h co-incubation on 48 hpf for cardiac and blood flow assays and 96 hpf zebrafish larvae for locomotion and oxygen consumption analysis. The test was conducted in duplicate.

## 3. Results

### 3.1. Total Distance Traveled for Zebrafish Larvae after RAC Exposure

Initially, we performed photomotor response (PMR) assay to evaluate the potential effects of RAC on zebrafish locomotor activity responding to four alternative dark/light cycle switches. After PMR assay, we extracted data and compared the total distance traveled, burst, and rotation counts of zebrafish larvae between the control and RAC exposed groups. The total distance chronology was measured on the basis of the velocity range of our previous publication [29,31]. Basically, zebrafish embryos display higher mobility in the dark phase than light phase in the PMR assay, and this pattern was still well distributed even after RAC treatment (Appendix B
Figure A1). We observed RAC provoked significant increase (*p* < 0.05) of average total distance traveled at 0.1 ppm (7.367 ± 0.124 cm), 1 ppm (7.766 ± 0.161 cm), 2 ppm (7.370 ± 0.128 cm), and 8 ppm (8.477 ± 0.182 cm) concentration in the light cycle compared to the control (6.253 ± 0.096 cm) (Figure 1A and Appendix A). Also, in the dark cycle, zebrafish exposed to RAC 0.1 ppm (11.090 ± 0.135 cm), 1 ppm (11.660 ± 0.131 cm), 2 ppm (12.460 ± 0.118 cm), and 8 ppm (13.120 ± 0.185 cm) displayed higher average distance traveled than the control (8.880 ± 0.099 cm) (Figure 1B). One data point at 4 ppm, however, showing significant lower total distance traveled (*p* < 0.05) in both light (5.468 ± 0.163 cm) and dark cycle (8.668 ± 0.177 cm). This result reveals that the administration of RAC to zebrafish larvae can trigger psychoactive effects at both dark and light phases for the PMR assay.

### 3.2. Total Rotation Movement for Zebrafish Larvae after RAC Exposure

The rotational chronology (clockwise or anti-clockwise) was influenced by many discrete agents such as feeding, brightness, or external pressure [44]. In response to RAC exposure, similar with previous locomotor results, zebrafish larvae displayed increment of rotational movements at both light and dark cycles. Results of RAC exposure showed significant increased (*p* < 0.05) on zebrafish rotational movement activity during both light and dark cycles in the range from 1 to 8 ppm concentration (Figure 1C,D). However, the RAC 0.1 ppm group showed hypoactivity in the dark cycle rotation test. In conclusion, the administration of RAC to zebrafish embryos can trigger high rotation movement at both dark and light phases in the PMR assay.

### 3.3. Total Burst Count for Zebrafish Larvae after RAC Exposure

Total burst count was determined by calculating the rapid change in the body activity of an experimental animal with higher than 20 pixels per second, that is an effective and sensitive index in evaluating the irregular movement and anxiety in zebrafish [29]. Statistical analysis showed that RAC induced higher average burst movement in zebrafish larvae in light cycle at all treated concentrations i.e., 0.1 ppm (2.822 ± 0.129 burst count/min), 1 ppm (6.960 ± 0.262 burst count/min), 2 ppm (8.398 ± 0.286 burst count/min), 4 ppm (7.659 ± 0.262 burst count/min), and 8 ppm (5.732 ± 0.163 burst count/min) compared to the control group (1.946 ± 0.130 burst count/min) with *p* < 0.0001 (Figure 1E). In comparison to light phase, zebrafish larvae displayed more burst movement in dark phase at 0.1 ppm (19.840 ± 0.274) 1 ppm (31.270 ± 0.424 burst count/min), 2 ppm (36.610 ± 0.451 burst count/min), 4 ppm concentrations (24.750 ± 0.333 burst count/min), and 8 ppm (21.460 ± 0.298) comparing to the control (15.400 ± 0.368 burst count/min) (Figure 1F). In conclusion, the administration of RAC to zebrafish embryos can trigger high burst movement at both dark and light phases for the PMR assay.

### 3.4. Respiratory Rate Analysis in Response to RAC Exposure in Zebrafish Larvae

As previous locomotor data showed that RAC exposure to zebrafish larvae cause hyperactivity. This intriguing observation leads us to ask whether the locomotor hyperactivity is associated with oxygen consumption alteration. In order to validate this hypothesis, we conducted oxygen consumption measurement on zebrafish embryos after RAC treatment by using Loligo Microplate Respirometry System (Loligo Systems, Viborg, Denmark) (Figure 2A). When the blue LED irradiated on the oxygen sensitive sensor embedded in the microplate, the blue light energy will be absorbed by O_2_ and emitted as different intensity of red-light signals. By monitoring the relative oxygen level in the microplate over time, we found the dissolved oxygen level is sharply declined in embryos treated with RAC with significant *p* < 0.001 (Figure 2B). By statistical quantification, we identified a significant increase of oxygen consumption in zebrafish embryos after exposed to different concentrations of RAC starting at the lowest 0.1 ppm (3.081 ± 0.158 ppm), 4 ppm (3.258 ± 0.121 ppm), and to the highest 8 ppm (3.411 ± 0.144 ppm). This result supported our hypothesis that acute RAC exposure in zebrafish caused hyperactivity and associated with high oxygen consumption (Figure 2C).

### 3.5. Cardiovascular Performance Assay

Next, we performed several important assays to elucidate the corresponding alteration of cardiovascular system after RAC exposure in zebrafish embryos at 72 hpf after 24-h exposure to RAC from 48 hpf onwards. To measure cardiac performance, several cardiac physiology endpoints were analyzed, such as heart rate in atrium, heart rate in ventricle, stroke volume, cardiac output, ejection fraction, and shortening fraction. We also checked the cardiac rhythm by analyzing the heart rate variability in both atrium and ventricle chambers and time interval between atrium to ventricle relaxation and vice versa. Furthermore, blood flow velocity in dorsal aorta was analyzed to measure the effect of RAC exposure to vascular system in zebrafish.

Heart rates show the overall condition of the heart. Lower heart rate usually showed that the person has a weaker health condition. However, in some special case like athlete, they will have lower resting heart rate which come from the increase power of the heart muscle [45]. In contrary to the lower heart rate, a higher heart rate is not preferable as it in many cases related to stress conditions and other cardiovascular related diseases [46]. It is intriguing to find that although locomotor hyperactivity and high oxygen consumption were detected, heart rate still maintains a regular level after RAC exposure at RAC doses of 0.1 and 4 ppm (Figure 3A,B). At a high RAC dose of 8 ppm, significant heart rate elevation was detected in both atrium and ventricle chambers (Figure 3A,B).

Stroke volume showed the volume of blood that leaves the heart chamber for each contraction. Stroke volume was calculated by subtracting the heart volume at the end-diastolic phase by the end-systolic phase that assumed that the heart chamber is an ellipsoid shape [47]. After incubation in RAC, we observed significant elevation of stroke volume after incubation in 4 ppm (93.260 ± 8.070 pL/beat) and 8 ppm (100.800 ± 6.529 pL/beat) (Figure 3C), which suggested that RAC could increase the heart contractility.

Cardiac output is defined as the total volume of blood pumped from the ventricle per minute, which is calculated by multiplying the stroke volume with the heart rate. Just as with stroke volume result, significant elevation also observed in zebrafish cardiac output after 4 ppm (14805 ± 1437 pL/min) and 8 ppm (16618 ± 942 pL/min) incubation of RAC (Figure 3D). Those effects suggest the cardiac output elevation after RAC exposure is largely contributed by stroke volume rather than heart rate elevation.

The ejection fraction is the measurement that how much blood is pumped out of the ventricle and is calculated by dividing the stroke volume with heart volume at the end diastolic phase. Ejection fraction refers the ability of heart to pump blood [48]. Therefore, the more blood volume pumped out, the increased ability of heart when treated with 8 ppm RAC (45.94 ± 1.64 EF%) was displayed (Figure 3E). Shortening fraction is measured by calculating the changing percentage in the dimension of ventricle during systolic phase [49]. Shortening fraction is correlated with ejection fraction [50]. To pump a large volume of blood (stroke volume) with high force (Ejection fraction), the lost dimension and percentage changes in ventricular dimension occurred when treated with 4 ppm (16.86 ± 1.41 SF%) and 8 ppm RAC (18.42 ± 0.96 SF%) (Figure 3F).

Finally, we also measured heart rate variability to evaluate whether acute RAC exposure can trigger arrhythmia in zebrafish embryos or not. Furthermore, we also checked the time interval between atrium to ventricle muscle relaxation and vice versa. By using Poincare plot, sd1 and sd2 value showed no significant difference in both the atrium and ventricle chamber (Figure 4A–D). No significant change was observed in the time interval between each chamber beat in response to RAC exposure, which suggested that RAC exposure did not change the heart rate variability and rhythm of zebrafish larvae (Figure 4E,F).

### 3.6. Blood Flow Measurement after RAC Exposure

In this endpoint, we observed the blood flow velocity in the dorsal aorta of zebrafish in correlation with cardiac analysis (The measurement position was highlighted in Figure 5A,B). The blood flow images were captured by high-speed CCD setting and later the maximal and average blood flow velocity was analyzed by using ImageJ-based method according to our previously published protocol described by Santoso et al. [33]. Heart rate and blood flow are tightly associated each other and could be influenced by chemical exposure like RAC. Results show RAC at 4 ppm can significantly enhance the average and maximal blood flow velocity with statistical means ± SEM value of 448.70 ± 24.18 µm/s and 963.40 ± 65.73 µm/s, respectively (Figure 5C–E and Appendix A). Based on data collected from Figure 3 and Figure 4, we found RAC exposure affects neither heart rate nor cardiac rhythm, but indeed can influence the stroke volume, cardiac output and ejection, and shortening fractions in zebrafish embryos. This alteration triggered by RAC can enhance the pumping capacity and blood volume without effecting size of heart chamber and cardiac rhythm in zebrafish embryos. Relating our blood flow results with cardiac performance in zebrafish, we concluded that RAC can increase the blood flow velocity in zebrafish without increasing heart rate too much (comparing Figure 3A,B and Figure 5D,E).

### 3.7. Molecular Docking for RAC and Endogenous β-adrenergic Receptor

In zebrafish genome, 10 genes were annotated as β-adrenergic receptors. We tried to validate which receptor has the higher affinity with RAC and might play a role on mediating RAC physiological activity, also comparing with PROP (a β-adrenergic receptor blocker). First, we searched ZFIN database (https://zfin.org/; accessed on 14 June 2021) for gene annotated with adrenergic receptors and later download the amino acid sequences for structure fitting. Briefly, after homology modeling structures of zebrafish endogenous β-adrenergic receptor subtypes were made. The amino acid sequence (aa) of zebrafish (*Danio rerio*) ten subtypes were selected from the Uniprot database (https://www.uniprot.org/; accessed on 9 March 2021) and their corresponding IDs and chromosomal position were listed in Table 1. The template structures for modeling structures were picked by NCBI blastp with the four PDB crystal protein structures (with 29.66% to 69.15% protein sequence identities). Ultimately, the crystal structure, 6kux, is chosen because the 6kux ligand has the best overlap with RAC among the four ones. The MODELLER software v9.20 constructed all homology modeling structures for flexible molecular docking. CDOCKER scores were assigned by the molecular docking module, flexible CDOCKER, indicating the binding affinity between each target protein and RAC [36,37]. Results from molecular docking showed that RAC may be able to interact with ten subtypes of zebrafish adrenergic receptors similar with the human homolog with CDOCKER scores of either 47.62 (adra1aa), 44.77 (adra1ab), 42.85 (adra1ba), 44.83 (adra1bb), 42.50 (adra1d), 43.68 (adra2a), 46.91 (adra2b), 41.90 (adra2c), 44.27 (adra2da), 42.74 (adra2db), or 44.15 (adra2a, *Homo sapiens*). PROP interacts with ten subtypes of zebrafish adrenergic receptors with CDOCKER scores of either 25.37, 25.97 (adra1aa), 22.98, 22.40 (adra1ab), 22.51, 22.41 (adra1ba), 25.58, 27.33 (adra1bb), 23.17, 23.37 (adra1d), 21.53, 23.25 (adra2a), 24.93, 24.48 (adra2b), 24.20, 23.57 (adra2c), 24.42, 22.44 (adra2da), 25.17, 22.50 (adra2db), or 21.83, 20.62 (adra2a, *Homo sapiens*) in R form and S form, respectively. Based on in silico data, PROP overall has about 50% lower binding affinity than RAC. Take zebrafish adra1aa as an example, the endogenous β-adrenergic receptor binding pocket for RAC was identified in the middle of the hollow cylinder of the seven transmembrane domains by homology modeling (Figure 6A). Three-dimensional (3D) (Figure 6B,C) structures illustrate three hydrogen bond formations between RAC and zebrafish adra1aa at the positions Glu101, Ser418, and Asp120 (highlighted by the green and blue dotted line). In conclusion, based on molecular docking, RAC might be able to bind to the ten zebrafish β-adrenergic receptors subtypes, from adra1aa to adra2db, with strong binding affinity supported compared to their human homolog.

### 3.8. Test Physiological Effects of β-Blocker ‘Propranolol’

We also tested potential effects of PROP exposure on zebrafish larvae using different behavioral and physiological endpoints after 24-h incubation. PROP has been found to be effective in performance and somatic anxiety [51]. Our PROP data supports the previous study showing PROP failed to affect the anxiolytic activity in a specific manner and pattern [52]. At the highest tested dose of PROP (8 ppm), zebrafish larvae showed lowered total distance in both light/dark cycle and this pattern was the same in other endpoints like rotational and burst movement (data shown in Table A1). But other tested concentration exhibited an irregular pattern in locomotory behavior. This shows that PROP can be only effective at high dose to treat the acute anxiety disorders. Previous studies reported the PROP toxicity after chronic exposure of 14 days in Japanese Medaka resulted in reduced growth rate [53] and significant changes in plasma steroid level [54]. In vertebrates, β-adrenoreceptors are mainly located at cardiac and skeletal muscles [55] and are also found in fish myocardium [56].

In our study, PROP exposure can significantly lower the heart rate at 4 and 8 ppm concentrations (for both atrium and ventricle chambers, Table A2) support previous literature show PROP lowered heart rate in zebrafish larvae [53,57]. Interestingly, we did not observe any significant alteration in average and maximum blood flow velocity in zebrafish larvae exposed to different doses of PROP (Table A2) despite lowered atrium and ventricle heart rate was detected. In addition, oxygen utilization was significantly higher at PROP 8 ppm dose (Table A2) as that of RAC. As our molecular docking data demonstrates that PROP has less binding affinity than RAC (Figure 6 and Table 1). Previous literature states that PROP has been found to treat cardiovascular diseases and lowering the heart rate [8,58] and β-adrenoreceptors are primarily located in cardiac tissues [55]. Basically, our cardiac results from PROP exposed zebrafish larvae support above previous studies. However, PROP failed to exhibit and set any specific pattern for other behavioral and physiological parameters we studied. Further acute and chronic studies of PROP exposure could help to better understand the effect on behavioral and physiological parameters.

### 3.9. Rescue Physiological Alterations Induced by RAC with PROP

β-blockers are extensively used for the treatment of cardiovascular diseases in human like hypertension, heart failure, arrhythmia, angina, migraines, and other related conditions [8,58,59,60], which works by competitive inhibition of β-adrenergic receptors. One of these drugs is PROP, which is being extensively used since its discovery in 1960s [61]. Considering the previous literature [62], we tested two PROP doses (1 and 4 ppm) at our RAC highest tested dose (8 ppm) and analyzed different behavioral and physiological endpoints like locomotion, oxygen consumption, cardiac physiology, and blood flow.

As a result, co-incubation of RAC and PROP indeed can significantly reduce locomotion hyperactivity triggered by 8 ppm RAC. The total distance travelled in both light and dark cycles for RAC exposed fish larvae can be significantly reduced after receiving either 1 or 4 ppm PROP administration (Figure 7A,B). For rotation behavior, we found PROP exposure also can effectively rescue high rotation behavior triggered by RAC in both light and dark cycles (Figure 7C,D). For burst movement behavior, the relative high burst activity can effectively restore after PROP exposure in both light and dark cycles (Figure 7E,F). Taking together, all evidences collected from light–dark switching photomotor assay suggests RAC and PROP function by competing with β-adrenoreceptor in the opposite direction.

As per previous literature, β-blockers are primarily used for cardiovascular diseases on fighting against arrhythmia [8,58,60]. Co-incubation of PROP with RAC significantly reduced the tachycardia as per our hypothesis at highest tested concentration. Figure 8A,B shows tachycardia phenotype triggered by 8 ppm RAC exposure can be effectively rescued by either 1 or 4 ppm PROP administration. Other cardiac physiological endpoints like stroke volume, cardiac output, ejection and shortening fraction display no significant alterations after PROP co-incubation (Figure 8C–F). Like our previous results show in Figure 4, RAC exposure did not alter the cardiac rhythm endpoints i.e., A–V and V–A interval, SD 1 and SD 2 for either atrium or ventricle. In contrary, co-incubation of both RAC and PROP significantly altered the cardiac rhythm endpoints by showing more higher levels of SD1 and SD2 in atrium (Figure 8I,J). The SD1 and SD2 levels in ventricle, on the contrary, display no significant different on RAC–PROP exposure compared to control (Figure 8K,L).

Since cardiac function alterations are tightly associated with blood flow dynamics, we aimed to analyze possible blood flow alteration after PROP rescue in RAC exposed fish. We used the same protocol as mentioned in Section 3.6 for blood flow measurement. RAC significantly increased the maximum and average blood flow velocity at a 4 ppm dose as shown in Figure 5D,E. Co-incubating PROP with RAC, however, reduced the maximum and average blood flow velocity as regarding with only RAC exposure. Significant differences in maximum and average blood flow velocity after co-treated RAC with PROP 1 and 4 ppm were displayed in Figure 9A,B when compared to RAC only. As per our results from oxygen consumption after RAC exposure in zebrafish larvae, all of the tested concentrations consumed more oxygen than the control (Figure 2C). PROP exposure alone did not seem to be lowering the oxygen consumption rate in zebrafish larvae (Table A2). However, the RAC–PROP co-incubation groups displayed significant lower oxygen consumption level (*p* < 0.0001) compared to the RAC-treated group (Figure 9C). Further, co-incubation with PROP 1 ppm could lowered the oxygen uptake to similar level with control (*p* = 0.2658). Taking together, all these data support RAC and PROP act through β-adrenergic receptors on mediating blood flow control in the opposite direction.

## 4. Discussion

Previously, it was first reported that RAC was associated with arterial and cardiac damage in canine. Based on cardiac anatomy, apoptosis within myocyte was the key reason that showed heart failure due to RAC exposure [20]. However, there was lack of extensive investigation of mechanism when exposing RAC. Therefore, in this study, we demonstrated serial experiments that associated with locomotion, oxygen consumption, and cardiovascular physiology endpoints, which were dedicated to identify the possible physiological alterations by using zebrafish as a simple animal model. The most important finding for this paper is that we identified the RAC function as a highly potent chemical reagent to boost the cardiovascular, oxygen consumption and locomotor activities in zebrafish embryos. This finding basically fit with the typical function of RAC as metabolism modifier to promote the metabolic rate in livestock [9,63] and fish [23]. Further, we performed a rescue trial in response to RAC induced alterations, using PROP as β-blocker or β-adrenergic receptor antagonist, which is known as non-selective β-blocker agent and is chiral in nature, to validate our results. We also analyzed the effects of only PROP exposure on zebrafish larvae using locomotory, cardiac, oxygen consumption, and blood flow endpoints on different concentrations. By utilizing this combinational approach, we were able to validate the potential mechanism of RAC on mediating zebrafish physiological alterations.

Considering the behavioral effects of acute RAC exposure to zebrafish larvae, we found the hyperactivity (increased total distance traveled, total rotational movements, and total burst count) in zebrafish larvae, which is similar to some previous studies in different animal models including zebrafish larvae [64]. In pigs, RAC treated animals were found to be more active and alert just after the onset of RAC feeding [63], but after the third week there was no difference in pig behavior. RAC was also found to be interacting with stress in rats [65]. Different studies of zebrafish exposure to RAC have stated increased locomotor activity and exploratory behavior, showing hyperactivity in zebrafish larvae [23]. Another study demonstrates the unaltered behavior in adult zebrafish at highest tested dose [22]. Taken these studies together, it can be said that RAC affects locomotory behavior differently depending upon developmental stages of zebrafish. While co-incubation of RAC with PROP, significantly increased the anxiolytic behavior of different locomotory endpoints in zebrafish larvae at the highest tested dose, which supports the previous studies that PROP has a significant role in reducing anxiety and hypertension [66,67,68]. Compared to RAC 8 ppm, co-incubation of PROP with RAC significantly reduced different locomotory endpoints, especially for burst and rotation.

RAC exposure significantly affected the cardiac physiology endpoints, i.e., heart rate, stroke volume, cardiac output, ejection fraction, and shortening fraction without significantly altering the cardiac rhythm endpoints. Significant results of cardiac physiology endpoints relate the findings with increased blood flow in zebrafish larvae exposed to RAC. It is intriguing to notice the faster heart rate and blood flow in RAC exposed zebrafish embryos did not trigger any sign of heart rate irregularity. This result is consistent with previous findings reported in finishing steers [23,69]. While co-incubation of PROP with RAC, significantly reduced the heart rate in the atrium and ventricle, which confirms the previous studies and supports the working mechanism of PROP in reducing heart rate [53,57,62,70]. The co-incubation of RAC and PROP also can rescue the high blood velocity and high oxygen consumption rate triggered by RAC even PROP are given at low dose as 1 ppm.

The alteration effects showed in this study are mainly administered by RAC that served as β-adrenergic agonist. The β-adrenergic agonist has been previously reported to regulate multiple physiological and metabolic functions including muscle system, heart rate, oxygen uptake, and also hemoglobin–oxygen affinity in fish [31,53,71]. In this study, we provided evidence of RAC physiological effects using well-established animal model, zebrafish by using multiple parameters. In line with previous study, RAC is able to increase the distance traveled and heart rate in zebrafish embryo and larvae [23]. Further, the increased oxygen consumption in RAC treatment might be due to additional muscle mass accretion. The over-abundance of F-actin-capping protein, which involves in skeletal muscle organization steers the oxygen transportation acceleration [72]. It also has been reported increasing anxiety, aggressive behavior, blood pressure, and palpitation in rats and pigs fed RAC [63,65].

For mechanism study, we have done the molecular docking analysis to demonstrate RAC have generally strong binding affinity to ten zebrafish subtypes of endogenous β-adrenergic receptors comparable to a human subtype. Therefore, based on molecular docking, we proposed the high potency on elevating cardiovascular and locomotion physiology in zebrafish might be mediated by multiple β-adrenergic receptor activation. It has been reported that S-propranolol has more β-adrenergic blocking activity than R-propranolol in mammals [73]. However, our in silico molecular docking study shows that the zebrafish subtype, adra1aa, binding to PROP with a chiral center, gains the highest CDOCKER scores for S form, 27.33, while the human adra2a (6kux) gains 21.83 and 20.62, in R form and S form, respectively. PROP overall has lower binding affinity than RAC (Figure 6 and Table 1). According to zebrafish genetic database *adra1ba, adra1bb, adra2a, adra2b,* and *adra2db* are expressed in heart of zebrafish while *adra2a, adra2da,* and *adra2c* are expressed in CNS [74]. *adra1d* and *adra1ab* genes are responsible for blood circulation while *adra2da* is expressed in the musculature system [75]. Our molecular docking results show that adra1aa has the highest binding affinity (47.63) among all tested subtypes (Table 1), which in human is responsible for Alzheimer’s disease and hypertension [76].

Based on this data, we were intrigued to evaluate the competition effect of RAC and PROP. We believed despite PROP having a lower binding affinity than RAC, it might still be able to display an antagonistic effect to inhibit RAC. In accordance with a prior study, PROP as β-receptor antagonist responsible for the neutralization of RAC effects [77]. Co-incubation of PROP and RAC in the same doses prevented RAC-stimulated cAMP production in myotubes, which is highly related in skeletal muscle adaptation. Another study showed PROP co-administered with β-adrenergic agonist, clenbuterol, is able to block the rise in cardiac muscle growth response, but PROP itself did not inhibit the ability of clenbuterol to stimulate protein accretion [78]. As of yet, the exact explanation remains unclear. In this case, whether PROP is literally able to inhibit RAC directly or it acts by working through a different β-receptors may need further investigations. In the future, administration of β-adrenergic receptor subtype-specific antagonist or perform systematically gene knockout/knockdown experiment for zebrafish endogenous β-adrenergic receptors might be able to provide more direct evidence to elucidate this hypothesis. Literature also shows that β-adrenergic agonists are associated with increased cardiac rate in different animal species like dogs along with pigs and fish [20,79]. The persistent increased cardiac rate in response to β-adrenoceptor agonist can results in cardiomyocyte death due to inflammation and impaired calcium handling [80]. Further studies are required to explore the potential toxicity of chronic RAC exposure of zebrafish on cardiovascular physiology.

## Figures and Tables

**Figure 1 cells-10-02449-f001:**
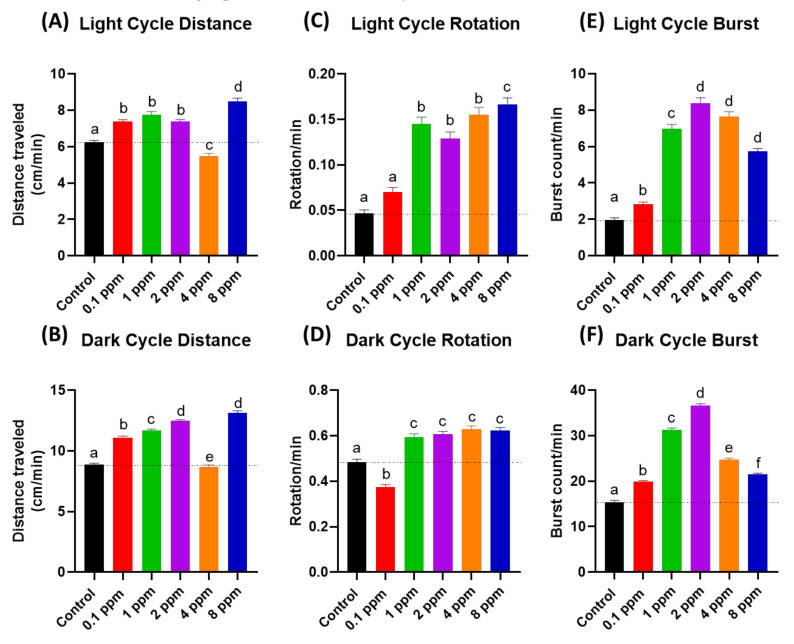
The RAC effect on zebrafish photomotor activities. (**A**) Total distance traveled of zebrafish larvae after 24-h incubation to different doses of RAC in (**A**) light cycle and (**B**) dark cycle. Total rotation movement of zebrafish larvae after 24-h incubation to different doses of RAC in (**C**) light cycle and (**D**) dark cycle. Total burst movement of zebrafish larvae after exposed to different doses of RAC for 24 h in (**E**) light cycle and (**F**) dark cycle. The data were shown as the Mean ± SEM and analyzed by Kruskal–Wallis test followed with Dunn’s multiple comparisons test. Different letters a, b, c, d, e, and f above columns indicate significant statistical differences with *p* < 0.05 (*n* = 96 for all groups). Black bar: control, red bar: RAC 0.1 ppm, green bar: RAC 1 ppm, purple bar: RAC 2 ppm, orange bar: RAC 4 ppm, and blue bar: RAC 8 ppm.

**Figure 2 cells-10-02449-f002:**
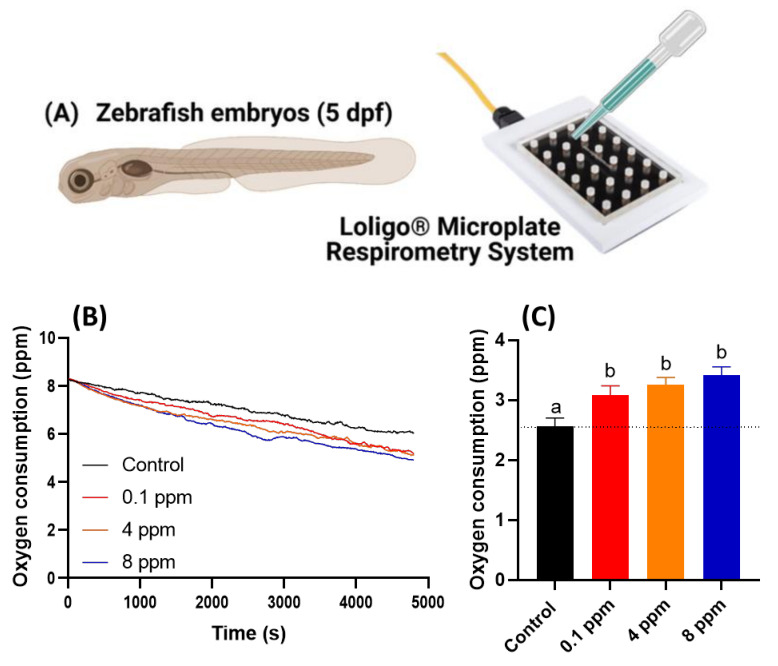
Oxygen consumption of zebrafish larvae at 5 dpf after 24-h incubation in different doses of RAC. (**A**) Experiment setting used to oxygen consumption measurement by using commercial instrument of Loligo Microplate Respirometry System. (**B**) The time chronology of dissolved oxygen level measured in the microplate. (**C**) The relative oxygen consumption for zebrafish embryos treated with different dose of RAC. The data are shown as the means ± SEM and analyzed by Two-way ANOVA either with Kruskal–Wallis One-way ANOVA test with Dunn’s correction as post-hoc multiple comparison test (*n* = 43–46). Different letters a and b above columns indicate significant statistical differences with *p* < 0.05. Black bar: control, red bar: RAC 0.1 ppm, orange bar: RAC 4 ppm, and blue bar: RAC 8 ppm.

**Figure 3 cells-10-02449-f003:**
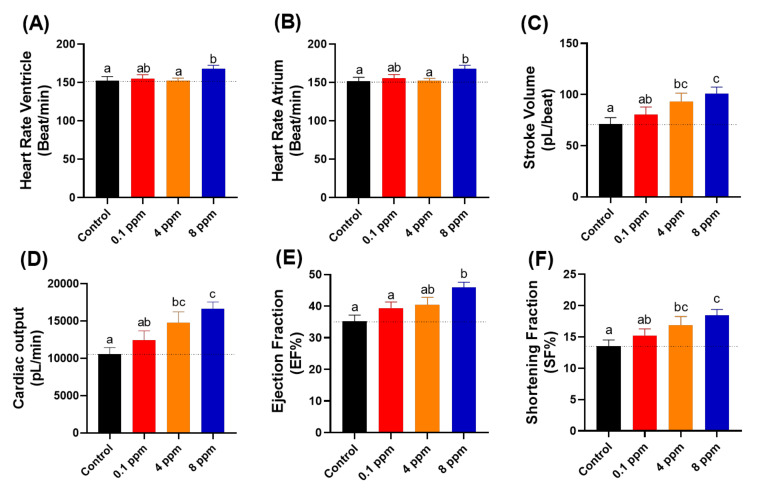
Cardiac physiology analysis of zebrafish larvae at 72 hpf after exposed to different doses of RAC. Several important parameters like (**A**) heart rate in the ventricle, (**B**) heart rate in the atrium, (**C**) stroke volume, (**D**) cardiac output, (**E**) ejection fraction and (**F**) shortening fraction were measured and compared. The data are expressed as the means ± SEM and were analyzed by one-way ANOVA test followed with Fisher’s LSD test. Different letters a, b, and c above columns indicate significant statistical differences with *p* < 0.05 (*n* = 27 for all groups). Black bar: control, red bar: RAC 0.1 ppm, orange bar: RAC 4 ppm, and blue bar: RAC 8 ppm.

**Figure 4 cells-10-02449-f004:**
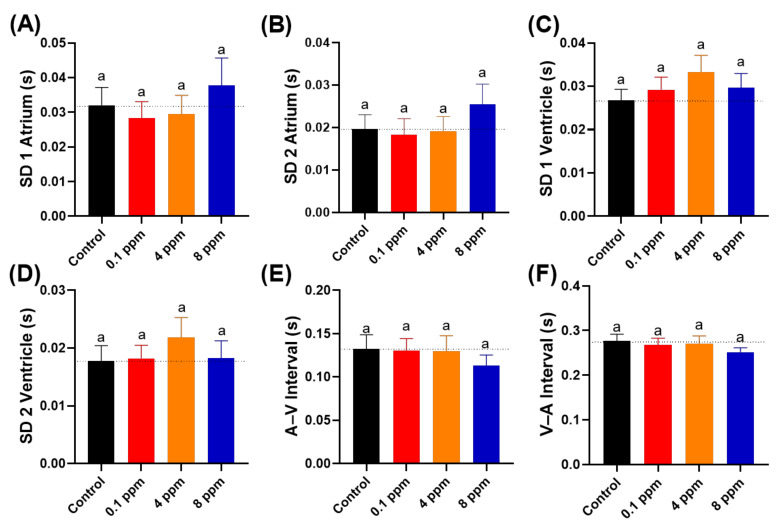
Cardiac rhythm analysis of zebrafish larvae at 72 hpf after 24-h exposure to different doses of RAC. Several important parameters like (**A**) SD1 of Poincare plot in atrium, (**B**) SD2 of Poincare plot in atrium, (**C**) SD1 of Poincare plot in ventricle, (**D**) SD2 of Poincare plot in ventricle, (**E**) A–V Interval and (**F**) V–A Interval were measured and compared. The data are expressed as the means ± SEM and were analyzed by Kruskal–Wallis test followed with Dunn’s multiple comparisons test (*n* = 27). Same letter ‘a’ above columns indicate no significant statistical differences with *p* > 0.05. Black bar: control, red bar: RAC 0.1 ppm, orange bar: RAC 4 ppm, and blue bar: RAC 8 ppm.

**Figure 5 cells-10-02449-f005:**
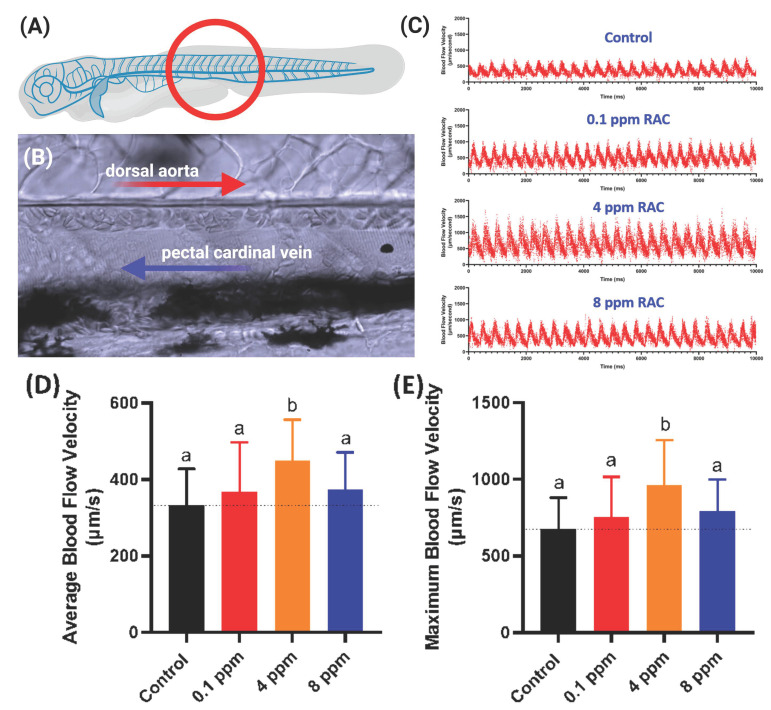
Evaluation of blood flow velocity in zebrafish larvae after exposed to RAC. (**A**) Schematic diagram showing the position for image capture to calculate blood flow velocity. (**B**) Magnification of red circle in (**A**) to show the position of dorsal aorta and post cardinal vein. (**C**) The time chronology of blood flow rate monitoring in the dorsal aorta of zebrafish embryos. (**D**) Average and (**E**) maximal blood flow velocity in the dorsal aorta of zebrafish larvae aged at 72-h post fertilization (hpf) after 24 h incubations in RAC from 48 hpf onwards. The data are shown as the means ± SD and were analyzed by one-way ANOVA test followed with Fisher’s LSD test. Different letters a and b above columns indicate significant statistical differences with *p* < 0.05 (*n* = 21 for control, *n* = 20 for all RAC groups). Black bar: control, red bar: RAC 0.1 ppm, orange bar: RAC 4 ppm, and blue bar: RAC 8 ppm.

**Figure 6 cells-10-02449-f006:**
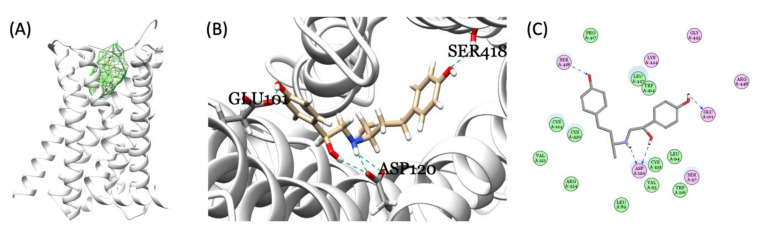
Molecular docking of RAC and zebrafish endogenous β-adrenergic receptor. (**A**) The identified binding pocket in the middle of the hollow cylinder of the seven transmembrane domain sub-types homology modeling structures of β-adrenergic receptor (adra1aa as an example). Three-dimensional (3D) (**B**) and 2D (**C**) illustrations of interactions between RAC and endogenous zebrafish β-adrenergic receptor showing the three hydrogen bond formations between RAC and adra1aa at the positions Glu101, Ser418, and Asp120 (highlighted by the green and blue dotted line).

**Figure 7 cells-10-02449-f007:**
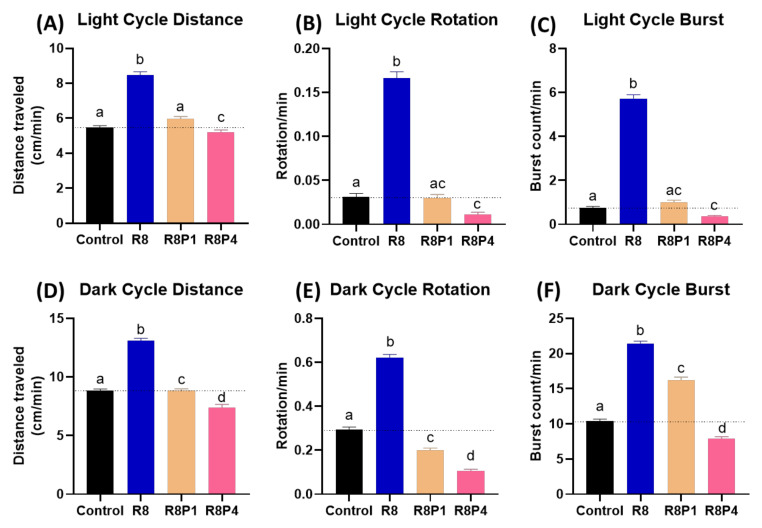
Locomotory activity evaluation after co-incubation of different doses of RAC and PROP in zebrafish larvae after 24-h incubation. Totally, six endpoints of (**A**) Light cycle distance, (**B**) Dark cycle distance, (**C**) Light cycle rotation, (**D**) Dark cycle rotation, (**E**) Light cycle burst, and (**F**) Dark cycle burst were measured and compared. The statistics were demonstrated as the mean ± SEM and analyzed by Kruskal–Wallis test followed with Dunn’s multiple comparisons test. Different letters a, b, c, and d above columns indicate significant statistical differences with *p* < 0.05 (*n* = 48). Abbreviations: R8 (RAC 8 ppm), R8P1 (RAC 8 ppm and PROP 1 ppm), and R8P4 (RAC 8 ppm and PROP 4 ppm). Black bar: control, blue bar: R8, cream bar: R8P1, and pink bar: R8P4.

**Figure 8 cells-10-02449-f008:**
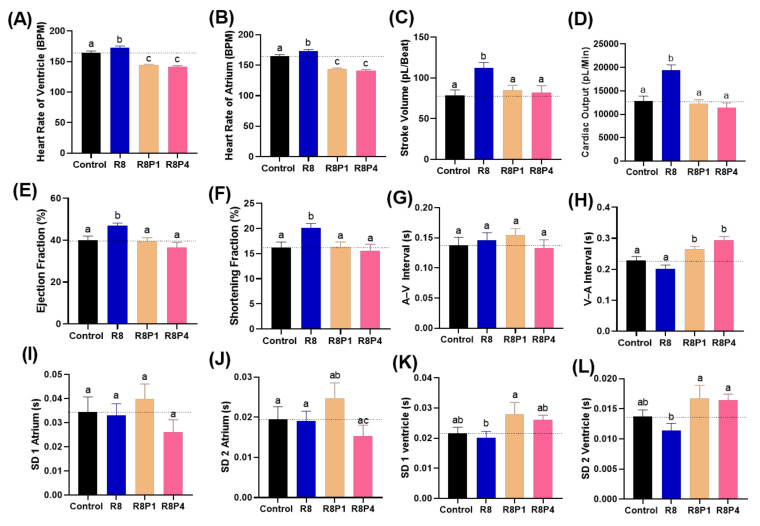
Cardiac physiology alterations of zebrafish larvae at 72 hpf after co-incubation of RAC and PROP at different doses. Several important endpoints like: (**A**) heart rate in the ventricle, (**B**) heart rate in the atrium, (**C**) stroke volume, (**D**) cardiac output, (**E**) ejection fraction and (**F**) shortening fraction (**G**) A–V interval, (**H**) V–A interval, (**I**) SD of Poincare plot in 1 atrium, (**J**) SD 2 of Poincare plot in atrium, (**K**) SD 1 of Poincare plot in ventricle, and (**L**) SD 2 of Poincare plot in ventricle were measured and compared. The data are expressed as the means ± SEM and were analyzed either by one-way ANOVA followed with Fisher’s LSD test or Kruskal–Wallis test with Dunnett’s correction for multiple comparison test. Different letters a, b, and c above columns indicate significant statistical differences with *p* < 0.05 (*n* = 20–22). Abbreviations: R8 (RAC 8 ppm), R8P1 (RAC 8 ppm and PROP 1 ppm), and R8P4 (RAC 8 ppm and PROP 4 ppm). Black bar: control, blue bar: R8, cream bar: R8P1, and pink bar: R8P4.

**Figure 9 cells-10-02449-f009:**
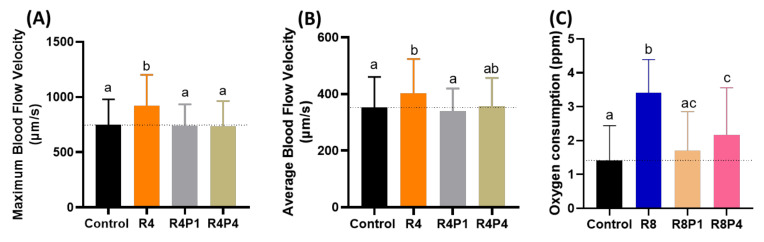
Evaluation of blood flow velocity and oxygen consumption in zebrafish larvae after co-incubating RAC and PROP. (**A**) Maximal and (**B**) average blood flow velocity in the dorsal aorta of zebrafish larvae aged at 72-h post fertilization (hpf) after 24 h incubations in RAC–PROP from 48 hpf onwards. The data are shown as the means ± SEM and were analyzed by One-way ANOVA test with Fisher’s LSD test (*n* Control = 41, *n* R4 = 40, *n* R4P1 = 20, and *n* R4P4 = 20). (**C**) Oxygen consumption of zebrafish larvae at 5 dpf after 24-h incubation in different doses of RAC and RAC-PROP combination. The data are shown as the means ± SD and analyzed by One-way ANOVA test with Dunn’s correction as post-hoc multiple comparison test (*n* = 42–46). Different letters a, b, and c above columns indicate significant statistical differences with *p* < 0.05. Abbreviations: R4 (RAC 4 ppm), R4P1 (RAC 4 ppm, PROP 1 ppm), R4P4 (RAC 4 ppm, PROP 4 ppm), R8 (RAC 8 ppm), R8P1 (RAC 8 ppm, PROP 1 ppm), and R8P4 (RAC 8 ppm, PROP 4 ppm). Black bar: control, orange bar: R4, silver bar: R4P1, gold bar: R4P4, blue bar: R8, cream bar: R8P1, and pink bar: R8P4.

**Table 1 cells-10-02449-t001:** Flexible molecular docking score for each homology modeling structure of endogenous β-adrenergic receptor subtype. The zebrafish subtype, adra1aa, binding to RAC, gains the highest CDOCKER scores, 47.63, while the human adra2a (6kux) gains 44.15. The zebrafish subtype, adra1aa, binding to PROP with a chiral center, gains the highest CDOCKER scores for S form, 27.33, while the human adra2a (6kux) gains 21.83 and 20.62, in R form and S form, respectively. PROP overall has lower binding affinity than RAC.

ZFIN Gene ID	Gene Name	Chromosomal Position	CDOCKER Score Ractopamine	CDOCKER Score Propranolol (R)	CDOCKER Score Propranolol (S)
ZDB-GENE-030131-2831	*adra1aa*	8	47.63	25.37	25.97
ZDB-GENE-060503-384	*adra1ab*	10	44.77	22.98	22.40
ZDB-GENE-120510-1	*adra1ba*	21	42.85	22.51	22.41
ZDB-GENE-041114-51	*adra1bb*	14	44.83	25.58	27.33
ZDB-GENE-090312-203	*adra1d*	1	42.50	23.17	23.37
ZDB-GENE-021010-1	*adra2a*	22	43.68	21.53	23.25
ZDB-GENE-021010-2	*adra2b*	8	46.91	24.93	24.48
ZDB-GENE-021010-3	*adra2c*	1	41.90	24.20	23.57
ZDB-GENE-021010-4	*adra2da*	14	44.27	24.42	22.44
ZDB-GENE-021010-5	*adra2db*	21	42.74	25.17	22.50

## Data Availability

The data presented in this study are available directed to the corresponding authors.

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
