# Peer review of "Evaluation of Effects of Ractopamine on Cardiovascular, Respiratory, and Locomotory Physiology in Animal Model Zebrafish Larvae"

_cells, 2021, doi:10.3390/cells10092449_

Round 1
Reviewer 1 Report
In this study, the investigators used zebrafish embryos to study the effects of ractopamine (RAC) on fish larval behavior. They focused on the physiological response with respect to locomotor activity, oxygen consumption and cardiovascular performance. They showed that RAC administration significantly enhanced locomotor activity, cardiac performance, oxygen consumption and blood flow rate in zebrafish embryos. In silico molecular docking analysis suggests ractopamine displays similar binding affinity to all ten subtypes of endogenous adrenergic receptors. The authors concluded that ractopamine might activate broad spectrum of β-adrenergic receptors to boost the locomotor activity, cardiac performance, and oxygen consumption in zebrafish.
Although increased locomotor activity has been reported in previous studies, this study provided more in-depth analyses of locomotor activity, oxygen consumption and cardiovascular performance upon RAC administration.
The in silico molecular docking analysis is informative, but very preliminary. Although ractopamine displays similar binding affinity to all ten subtypes of endogenous adrenergic receptors, it is not clear how many of these receptors are expressed in zebrafish embryonic cardiac muscle and other tissues that may mediate RAC action. Additional expression analysis and discussion could provide useful information on potential β-adrenergic receptors involved in different aspects of RAC action.
The authors suggest that RAC might play as high potent and broad spectrum β-adrenergic receptors agonist on boosting the locomotor activity, cardiac performance, and oxygen consumption in zebrafish. Are β-adrenergic receptors antagonists available? If antagonists are applied together with RAC, does this block RAC induced action?
Author Response
In this study, the investigators used zebrafish embryos to study the effects of ractopamine (RAC) on fish larval behavior. They focused on the physiological response with respect to locomotor activity, oxygen consumption and cardiovascular performance. They showed that RAC administration significantly enhanced locomotor activity, cardiac performance, oxygen consumption and blood flow rate in zebrafish embryos. In silico molecular docking analysis suggests ractopamine displays similar binding affinity to all ten subtypes of endogenous adrenergic receptors. The authors concluded that ractopamine might activate broad spectrum of β-adrenergic receptors to boost the locomotor activity, cardiac performance, and oxygen consumption in zebrafish. Although increased locomotor activity has been reported in previous studies, this study provided more in-depth analyses of locomotor activity, oxygen consumption and cardiovascular performance upon RAC administration. The in silico molecular docking analysis is informative, but very preliminary. Although ractopamine displays similar binding affinity to all ten subtypes of endogenous adrenergic receptors, it is not clear how many of these receptors are expressed in zebrafish embryonic cardiac muscle and other tissues that may mediate RAC action. Additional expression analysis and discussion could provide useful information on potential β-adrenergic receptors involved in different aspects of RAC action.
Eternally grateful for your valuable comments and suggestions. Thus, the authors added more discussion for better understanding about β-adrenergic receptors and RAC action in this revised manuscript (line 620-625). According to zebrafish genetic database; adra1ba, adra1bb, adra2a, adra2b, adra2db are expressed in heart in zebrafish while adra2a, adra2da, adra2c are expressed in CNS. adra1d, adra1ab genes are responsible for blood circulation while adra2da is expressed in musculature system.
Wang, Z., Nishimura, Y., Shimada, Y., Umemoto, N., Hirano, M., Zang, L., Oka, T., Sakamoto, C., Kuroyanagi, J. and Tanaka, T., 2009. Zebrafish β-adrenergic receptor mRNA expression and control of pigmentation. Gene, 446(1), pp.18-27.
The authors suggest that RAC might play as high potent and broad spectrum β-adrenergic receptors agonist on boosting the locomotor activity, cardiac performance, and oxygen consumption in zebrafish. Are β-adrenergic receptors antagonists available? If antagonists are applied together with RAC, does this block RAC induced action?
Thanks a lot for your constructive feedback. The authors also agree that the addition of extra experiment using β-blocker to prove our hypothesis was needed. For extra experiment, we added more data using propranolol (PROP) as one non-selective β-blocker. All the experiments were conducted with same protocol. The concentration of PROP was selected based on our initial testing with PROP alone (0.1, 4, and 8 ppm) in all assays. After co-incubating both chemicals, we have got significant decrease in locomotor (Figure 7, line 479-503) and cardiac physiology endpoints (Figure 8, line 504-525). So, it means that PROP tends to normalize the RAC action. The blood flow velocity result also demonstrates the decrease in velocity after co-incubation of RAC & PROP, as we got statistically significant results (Figure 9, line 526-550). The other endpoint, oxygen consumption parameters also displayed significant lower oxygen uptake after co-incubation treatment of RAC & PROP. Those extra data have been added in the updated manuscript to provide more sloid evidences.
Reviewer 2 Report
The paper of Abbas and coworkers aims to assess the effect of Ractopamine (RAC) on the locomotion, cardiovascular behavior and oxygen consumption of zebrafish larvae.
The manuscript is not completely original because the paper of Garbinato et al., 2020 reports the effects of RAC on the embryos and larvae of zebrafish as well even if with different timing of treatment.
The most interesting result is that related to molecular docking suggesting the possible binding of RAC with the beta-adrenergic receptor. However, it is only in silico and it is not experimentally validated.
I think that the discussion and presentation of the data should be revised trying to be more detailed and trying to highlight the data obtained and the most interesting outcomes in light of the data shown by other authors in zebrafish embryos/larvae/adults and other model organisms.
The authors have not considered the possibility that methylene blue used to maintain zebrafish larvae could interact with RAC. I think it is better to show as supplementary files some data of the results obtained by using free osmosis water or fish water.
On the other hand, the difference obtained by other authors on zebrafish could be related to different protocols used.
Moreover, it is not clear how many replicates with 50 animals have been performed? Replicates are really important with these types of experiments. Also because there is great variability and not a trend between the different concentrations of RAC used.
For example: why 4 ppm has different behaviour in total distance traveled of zebrafish larvae in respect to other concentrations (0,1 ppm-8 ppm). It is possible that with replicates the results will change.
The authors must add some videos showing for example some representative experiments for blood flow behaviour or locomotor activity.
Minor revisions are:
Check references: duplicates are present: see Garbinato et al., 2020.
Author Response
The paper of Abbas and coworkers aims to assess the effect of Ractopamine (RAC) on the locomotion, cardiovascular behavior and oxygen consumption of zebrafish larvae. The manuscript is not completely original because the paper of Garbinato et al., 2020 reports the effects of RAC on the embryos and larvae of zebrafish as well even if with different timing of treatment. The most interesting result is that related to molecular docking suggesting the possible binding of RAC with the beta-adrenergic receptor. However, it is only in silico and it is not experimentally validated. I think that the discussion and presentation of the data should be revised trying to be more detailed and trying to highlight the data obtained and the most interesting outcomes in light of the data shown by other authors in zebrafish embryos/larvae/adults and other model organisms.
The authors appreciate the reviewer’s suggestion. The authors had tried their best to improve the results and discussion part (line 564-568, 597-609, 616-636). Some essential information was also included regarding the new additional data. The data presentation was changed to perform all pairwise comparisons between each group means in order to display more detailed effect of RAC exposure in different concentrations. The supplementary (Appendix) data have also been added to the manuscript to further supporting the outcomes of our study (line 665-681).
The authors have not considered the possibility that methylene blue used to maintain zebrafish larvae could interact with RAC. I think it is better to show as supplementary files some data of the results obtained by using free osmosis water or fish water.
Thank you so much for pointing the confusion. We added methylene blue to initially prevent the fungal infection after we harvested the eggs and kept them until 2dpf in fish water by changing the medium daily. At the time of zebrafish larvae exposure to RAC, we changed the medium by ddH2O by keeping control and all treatment groups at same conditions. So, zebrafish larvae were not exposed to RAC in methylene blue.
On the other hand, the difference obtained by other authors on zebrafish could be related to different protocols used.
Thank you for pointing this matter. Actually, the protocols that were used in this study have been published and been already applied in our previous studies and successfully displayed good performance and reliable results. Similar results also displayed in other studies that show RAC caused an increase of heart rate performance and hyperactivity behavior.
Hussain, A.; Audira, G.; Malhotra, N.; Uapipatanakul, B.; Chen, J.-R.; Lai, Y.-H.; Huang, J.-C.; Chen, K.H.-C.; Lai, H.-T.; Hsiao, C.-D.J.B. Multiple Screening of Pesticides Toxicity in Zebrafish and Daphnia Based on Locomotor Activity Alterations. 2020, 10, 1224.
Santoso, F.; Sampurna, B.P.; Lai, Y.-H.; Liang, S.-T.; Hao, E.; Chen, J.-R.; Hsiao, C.-D. Development of a simple imagej-based method for dynamic blood flow tracking in zebrafish embryos and its application in drug toxicity evaluation. Inventions 2019, 4, 65.
Garbinato, C.; Schneider, S.E.; Sachett, A.; Decui, L.; Conterato, G.M.; Müller, L.G.; Siebel, A.M. Exposure to ractopamine hydrochloride induces changes in heart rate and behavior in zebrafish embryos and larvae. Environmental Science and Pollution Research 2020, 27, 21468-21475.
Moreover, it is not clear how many replicates with 50 animals have been performed? Replicates are really important with these types of experiments. Also because there is great variability and not a trend between the different concentrations of RAC used. For example: why 4 ppm has different behaviour in total distance traveled of zebrafish larvae in respect to other concentrations (0,1 ppm-8 ppm). It is possible that with replicates the results will change.
Thanks for your valuable comments. Yes, authors agree that replication tends to make results more clear and supportive. We have performed duplication and manuscript has been updated accordingly. All the tests were performed in duplicate with total 96 fish for each group in locomotor activity test, 46 fish for oxygen consumption test, 27 fish for cardiac performance test, and 20 fish for blood flow velocity test. The all graphic data have also been updated in this revised manuscript.
The authors must add some videos showing for example some representative experiments for blood flow behaviour or locomotor activity.
Thanks for your suggestion. Yes, adding some representative videos will be more helpful. We have attached two supplementary videos in this revised manuscript to show the locomotion and blood flow alterations after exposed to ractopamine.
Minor revisions are: Check references: duplicates are present: see Garbinato et al., 2020.
Pardon for the inconvenience. We have made the relevant corrections in reference section. Thank you for your correction.
Round 2
Reviewer 1 Report
None
Reviewer 2 Report
The authors responded to my comments and revised the manuscript appropriately.